# Machine Learning-Based Modelling and Meta-Heuristic-Based Optimization of Specific Tool Wear and Surface Roughness in the Milling Process

Siamak Pedrammehr [1,2,*], Mahsa Hejazian [3], Mohammad Reza Chalak Qazani [2], Hadi Parvaz [4], Sajjad Pakzad [1], Mir Mohammad Ettefagh [3] and Adeel H. Suhail [5]

1. Faculty of Design, Tabriz Islamic Art University, Tabriz 5164736931, Iran
2. Institute for Intelligent Systems Research and Innovation, Deakin University, Waurn Ponds, VIC 3216, Australia
3. Faculty of Mechanical Engineering, University of Tabriz, Tabriz 5166616471, Iran
4. Faculty of Mechanical and Mechatronics Engineering, Shahrood University of Technology, Shahrood 36199-95161, Iran
5. Department of Mechanical Engineering, Middle East College, PB 79, Al Rusayl, Muscat PC 124, Oman
* Correspondence: s.pedrammehr@gmail.com

**Abstract:** The purpose of this research is to investigate different milling parameters for optimization to achieve the maximum rate of material removal with the minimum tool wear and surface roughness. In this study, a tool wear factor is specified to investigate tool wear parameters and the amount of material removed during machining, simultaneously. The second output parameter is surface roughness. The DOE technique is used to design the experiments and applied to the milling machine. The practical data is used to develop different mathematical models. In addition, a single-objective genetic algorithm (GA) is applied to numerate the optimal hyperparameters of the proposed adaptive network-based fuzzy inference system (ANFIS) to achieve the best possible efficiency. Afterwards, the multi-objective GA is employed to extract the optimum cutting parameters to reach the specified tool wear and the least surface roughness. The proposed method is developed under MATLAB using the practically extracted dataset and neural network. The optimization results revealed that optimum values for feed rate, cutting speed, and depth of cut vary from 252.6 to 256.9 (m/min), 0.1005 to 0.1431 (mm/rev·tooth), and from 1.2735 to 1.3108 (mm), respectively.

**Keywords:** milling process; machine learning; meta-heuristic optimization; surface roughness; tool wear

**MSC:** 49K99

## 1. Introduction

The estimation of tool life is an important area of research in the field of material machining. In order to maximize tool life, the right set of parameters must be used to optimize and better use the cutting tool and machining process. The capacity of the cutting tools is being improved and is the subject of intensive study. The main goal of this experimental research is to investigate how the milling operation's cutting conditions—such as cutting depth, speed, and feed—affect tool wear.

The replacement of tools, which is a result of tool wear, is one of the most significant economic components of machining. The productivity of the process is often immediately impacted by a reduction in the material removal rate as the tool wear criteria decline. Surface roughness, on the other hand, must also be taken into account as a quality attribute of the machined surface. There are many different optimization techniques available to solve engineering problems such as this. Many academics in the past focused more on the machining process and paid less attention to milling operations, basing their studies

on modeling, tool wear, and surface roughness. Additionally, they did not assess their optimization strategy against those of other scholars.

A hybrid optimization approach was employed by Yildiz et al. [1]. There were comparisons made between their findings and those of other researchers to indicate that their strategy was effective in improving process parameters during machine operations using the Nelder–Mead local search algorithm and the optimization algorithm of Harris Hawks. In a separate study, they optimized the processing parameters for several industrial processes using the multi-verse, grasshopper, and Harris Hawks optimization algorithms [2]. Savkovic et al. [3] constructed trustworthy intelligent models for selecting the milling process's output features based on the machining process's input parameters. The ideal mixture of process factors for optimal response—surface roughness, material removal rate, and strength—was chosen by Khawaja et al. [4] after considering mathematical models and using the response surface approach. Chien and Tsai [5] also created a model for forecasting tool flank wear using the modeling approach. They then used a genetic algorithm to improve the model to identify the ideal cutting conditions for the machining of 17-4PH stainless steel. By looking into flank wear and surface roughness while turning an Al/SiCp metal matrix composite, Sahoo and Pradhan [6] looked into the machining properties. They used the Taguchi technique to do this. The outcome of their research demonstrates that adhesion and abrasion are the most frequent wear processes. The Taguchi approach was also used by Mia et al. [7]. Their experiment was designed using a Taguchi orthogonal array and a signal-to-noise ratio optimization method. They used coated carbide tools for rough machining and MQL to assess the tool's wear characteristics and surface roughness. The findings indicate that although cutting depth affects tool wear, cutting speed substantially impacts surface roughness. Tsao [8] further used the Taguchi approach to enhance milling process parameters. Based on the findings, the flank wear was reduced by 62 percent using the suggested strategy. A distinct approach is used by Amouzgar et al. [9] to optimize the depth of tool wear. They used evolutionary algorithms to reduce tool wear and the interface temperature after simulating a turning operation using the finite element approach first. Seventy percent of the calculation time was shaved off. Aluminum alloy 7075 was the choice of Savkovi et al. [10], who looked at how cutting forces and factors affected surface roughness during milling. As a consequence of their research, the cutting force was shown to be ideal when the surface area was kept to a minimum for all input values. Additionally, the best alternatives for creating moderate computational roughness were minimal values for cutting speed, feed per tooth, cutting depth, and average level.

Recently, deep learning and machine learning techniques have been developed and have gained a remarkable interest across different research fields such as materials, microstructure, manufacturing, and energy [11–13]. The data-driven modeling methods are not based on the physics of process, and they often have a higher accuracy in comparison with the classical models [14]. Jia et al. [15] employed the particle swarm optimization method to obtain the optimum parameters for the production of mechanical materials. Pilania et al. [16] made materials property prediction using kernel ridge regression. Hwang et al. [17] proposed a four-layer perceptron for the rolled steel bar mechanical properties. Chen and Gu [18] reviewed the implemented machine learning techniques in composite material modelling and designing, and the mechanical properties of cold-rolled strip have been investigated through the feedforward neural network by Lalam et al. [19]. The developed model predicts the yielding strength and ultimate tensile strength with $\pm 10$ (MPa) accuracy for 90 percent of the data. Xiong et al. [20] investigated the estimation of the steels mechanical properties using four machine learning methods including random forest, linear least-square, k-nearest neighbors, and multilayer perceptron. Xie et al. [21] employed machine learning method to develop the mechanical properties online prediction for the hot rolled steel plate. Rajesh et al. [22] studied machine learning methods that have been used in manufacturing processes. In addition, machine learning methods have been applied to optimize different milling parameters such as tool condition monitoring [23,24], chatter

detection and stability [25,26], prediction of cutting forces and power [27,28], and surface prediction [29,30].

In this research, the specific tool wear is introduced as a new output parameter named in which the material removal rate and tool wear are evaluated. The second target output is surface roughness. In order to reduce tool wear, increase surface smoothness, and increase the rate of material removal in an AISI 1045 alloy steel mill, three alternative optimization techniques have been put forward and contrasted. The main contribution of this research is to extract the optimal milling process parameters, including cutting speed (m/min), feed rate (mm/rev·tooth), and depth of cut (mm), in order to reach the optimal specific tool wear (mm) and surface roughness (cm/(cm$^3$/s)) using multi-objective GA. It should be noted that higher specific tool wear means lower surface roughness, and there is a need of Pareto front to extract the optimal solutions of the milling process. In addition, the multi-objective functions of the milling process are extracted using adaptive network-based fuzzy inference system (ANFIS) to imitate the process based on the results of experiment. The optimal solutions are recommended at the end of this study.

## 2. Materials and Methods

The initial step in the investigation was to choose a subject matter for study. The intended piece of work was constructed using AISI 1045 alloy steel blocks (example dimensions: $150 \times 80 \times 60$ (mm)). The tensile strength of this steel ranges from 570 to 700 (MPa), with low strain hardenability. In many different industries, axles, pins, belts, gears, pumps, and shafts are made from this normalized and softened alloy, which has a hardness range of 170 to 210 (HRC). A universal milling machine called the FP4MD was used for the experiments. With a spin speed range of 50 to 2500 (rpm) and a desk feed rate range of 0 to 900 (mm/min), this machine provides a positioning precision of $+/-0.005$ (mm). A 4-fluted face milling cutter, 80 (mm) in diameter, was utilized in the milling operation. An insert made of cemented carbide that was covered with TiN made up the cutting tool (ISO R245-12 T3 M-PM 4020). This insert may be used in semi-dry or dry machining operations because of the thermal shock resilience of the substrate and the PVD coating. The inserts utilized in the studies had a 13.4 (mm) engraved circular diameter, a 10 (mm) effective cutting-edge length, and a 1.5 (mm) corner radius.

This paper aims to extract the optimal block cutting parameters in progress to reach the optimum parameters of the process with the fewer testing trials (best surface roughness using less cutting force). One of the accurate machine learning techniques (ANFIS) was employed to extract the model to connect the cutting parameters to surface roughness and milling force. The hyperparameters of ANFIS were extracted using GA to reach the highest performance of the developed model. In this study, there were 2 different outputs (surface roughness and tool wear) for a dataset. The model was trained ten times to extract the optimal ANFIS model. The recorded tool wear factor and surface roughness were considered to develop the ANFIS model.

Figure 1 illustrates the scheme of the developed method in the current study as the main contribution, which combines ANFIS and the GA. According to Figure 1a, the process is started by selecting the appropriate cutting parameters. The surface quality was then evaluated using a roughness meter produced by the Taylor Habson firm after the FP4MD universal milling machine was finished. The Olympus company's BX60 optical microscope, which has a 50× to 1000× magnification range, was also used.

The proposed algorithm includes the data pre-processing for eliminating the out-of-range data to enhance the accuracy of the algorithm. The single-objective GA was applied to choose the optimum ANFIS hyperparameters to achieve the least mean square error (MSE) between the predicted and actual data as it is shown in the left dashed line rectangular of Figure 1b.

The algorithm as initiated twice to extract ANFIS1 and ANFIS2 models to calculate the workpiece's tool wear and surface roughness, as shown in the right dashed-line rectangle of Figure 1b, respectively. Then, the multi-objective GA as employed to extract the optimal

cutting parameters of block in progress order to reach the most appropriate outcomes, including the specific tool wear and low surface roughness.

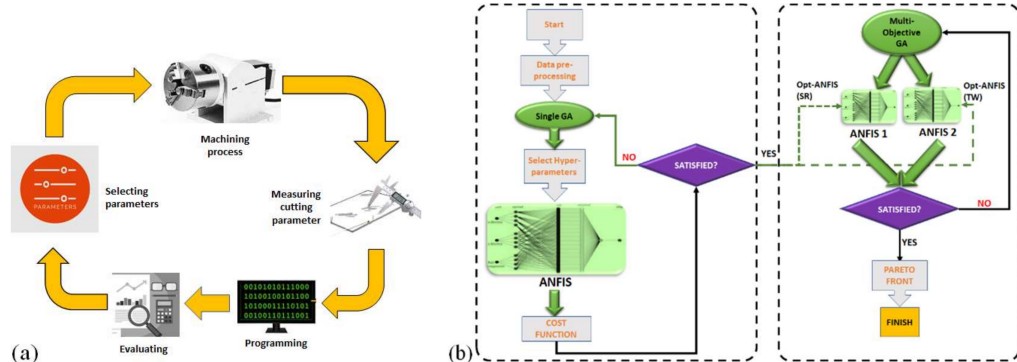

**Figure 1.** (**a**) The schematic structure of the whole proposed method; (**b**) the artificial intelligent section of this study.

The proposed algorithm includes the data pre-processing for eliminating the out-of-range data to enhance the accuracy of the algorithm. The single-objective GA was applied to choose the optimum ANFIS hyperparameters to achieve the least mean square error (MSE) between the predicted and actual data as it is shown in the left dashed-line rectangle of Figure 1b.

The algorithm was initiated twice to extract ANFIS1 and ANFIS2 models to calculate the workpiece's tool wear and surface roughness as shown in the right dashed-line rectangle of Figure 1b, respectively. Then, the multi-objective GA was employed to extract the optimal cutting parameters of block in progress order to reach the most appropriate outcomes, including the specific tool wear and low surface roughness.

Three variables: cutting speed, feed rate, and depth of cut, were considered in this research. Here, as the processes are time-consuming and expensive, out of 9 combinations, each with three replications, instead of 27 tests (full factorial) replications, the experiment was performed randomly by the Taguchi method ($L_9$).

## 3. Mathematical Modelling

The goal of a mathematical model is to build an input–output connection to explain how the dependent variable's usual value varies when each independent variable is changed. Furthermore, that is the initial stage to enable process optimization, the best milling settings to achieve the best possible surface smoothness, and particular tool wear. Table 1 displays the results of nine tests based on the Taguchi $L_9$ matrix.

According to the breakdown, the first three columns represent independent cutting parameters. In contrast, the fourth and fifth columns represent dependent and objective outputs representing surface roughness and tool wear.

**Table 1.** Practical test results with 9 replications based on Taguchi $L_9$ orthogonal array using cutting process parameters consisting of feed rate, cutting speed, and depth of cut with corresponding to the obtained specific tool wear and surface roughness.

|  | Cutting Speed (m/min) | Feed Rate (mm/rev·tooth) | Depth of Cut (mm) | SR (mm) | Specific Tool Wear cm/(cm³/s) |
|---|---|---|---|---|---|
| 1 |  | 0.06 | 1 | 1.67 | 0.0250 |
| 2 | 126 | 0.12 | 1.5 | 2.14 | 0.0106 |
| 3 |  | 0.18 | 2 | 2.22 | 0.0057 |
| 4 |  | 0.06 | 1.5 | 1.47 | 0.0139 |
| 5 | 201 | 0.12 | 2 | 2.04 | 0.0058 |
| 6 |  | 0.18 | 1 | 1.71 | 0.0075 |
| 7 |  | 0.06 | 2 | 1.75 | 0.0088 |
| 8 | 314 | 0.12 | 1 | 1.5 | 0.0077 |
| 9 |  | 0.18 | 1.5 | 1.94 | 0.0037 |

## 4. Optimization Procedure

Many real-world situations need the adjustment of process parameters to achieve desired results. A multi-target optimization issue aims to reduce two targets (responses) concurrently. As a result, one aim is often reduced while another is boosted in certain situations, which may lead to conflicting outcomes. If we want to find an acceptable solution that meets all of our goals while being as cost-effective as possible, we must use a trade-off approach. Tool wear during milling and workpiece surface roughness may be predicted using ANFIS, employed in this article to model these two variables.

### 4.1. Pre-Tuning of Algorithm

Three tasks need to be completed before creating the model and using the data. There are some statistics with a negative force at first. Since these data were produced as a result of the system's dysfunction, they ought to be deleted. Before using the data inside the models, the out-of-range data caused by a sensor malfunction should also be eliminated. In order to simplify the data for a system before training, the second stage involves normalizing or standardizing the procedure. Both approaches are used in this work to reduce the complexity of the network's input data and boost system accuracy. The standardization of the data is determined using the formula shown below:

$$\sigma_{x_i} = \frac{x_i - \overline{x}}{\sigma_x} \tag{1}$$

where $x_i$ and $\sigma_{xi}$ are, respectively, the raw and standardized $i^{th}$ inputs. $\sigma_x$ and $\overline{x}$ are also the functions to extract the standard and average deviations of the data. The normalized data is calculated as:

$$n_{x_i} = \frac{x_i - \underline{x}}{\overline{x} - \underline{x}} \tag{2}$$

### 4.2. Adaptive Network-Based Fuzzy Inference System

ANFIS is a hybrid system made up of an artificial neural network and a fuzzy inference system that is utilized as a kind of Takagi–Sugeno artificial intelligence model to resolve nonlinear and challenging situations (FIS). This suggests that ANFIS has the capacity for both self-learning and logical operation. The following are the ANFIS fuzzy rules:

- Rule 1: if t = O1, and x = P1:　　$z_1 = a_1t + a_2x + a_3$.
- Rule 2: if t = O2, and x = P2:　　$z_2 = b_1t + b_2x + b_3$.

In which $x$ and $t$ are inputs, $z_1$ and $z_2$ are the outputs specified by the mentioned rules. During the learning process, the parameters $a_1$, $a_2$, $a_3$, $b_1$, $b_2$, and $b_3$ are acquired.

The ANFIS structure shown in Figure 2 has five levels, each with an output and two inputs, as shown. Layers of meaning are as follows:

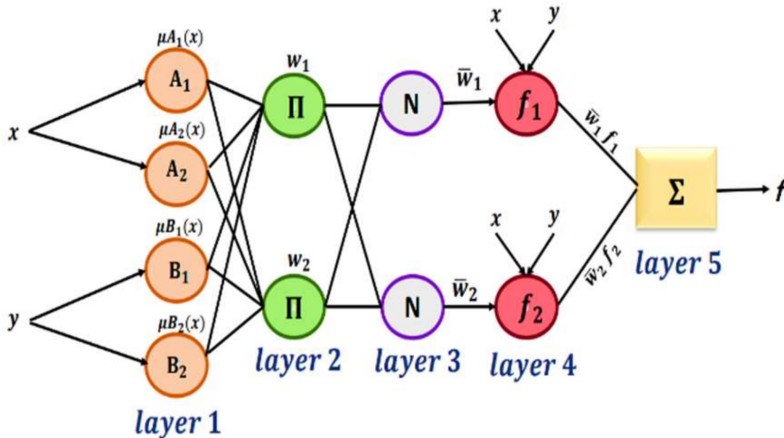

**Figure 2.** The structure of ANFIS.

In the first layer, membership functions are used to compute fuzzy clusters based on the training data input. The membership function is created using the indices ai and bi, and the degree of membership is calculated as follows:

$$Q_i^1 = \mu_{A_i}(x) = \frac{1}{1 + \left|\frac{x-c}{a}\right|^{2b}}; i = 1, 2 \tag{3}$$

$$Q_i^1 = \mu_{B_i}(x); i = 1, 2 \tag{4}$$

where $\mu_x$ and $\mu_y$ are the degrees of membership. $A_i$ and $B_i$ are fuzzy sets and $Q_i^1$ is the $i^{th}$ node in the $j^{th}$ layer output for input $x$.

Layer 2 is the rule that is applied to double the firing power. The output can be obtained as:

$$Q_i^2 = w_i = \mu_{A_i}(x) \cdot \mu_{B_i}(y), \ i = 1, 2 \tag{5}$$

Layer 3 used to normalize the firing intensity of the last layer. Dividing the $i^{th}$ rule's firing strength by the sum of all firing strengths, the normalized values can be obtained as:

$$Q_i^3 = \overline{w}_i = \frac{w_i}{w_1 + w_2 + w_3 + w_4}, \ i = 1, \dots, 4 \tag{6}$$

The defuzzification layer is the fourth layer. The result can be obtained by the dot product of the parameters set ($p_i$, $q_i$, and $r_i$) and the normalized firing strength, which yields:

$$Q_i^4 = \overline{w}_i f_i = \overline{w}_i (p_i x + q_i y + r_i) \tag{7}$$

Adding each rule's defuzzification outputs creates the final result of layer five:

$$Q_i^5 = \sum_i \overline{w}_i f_i = \frac{\sum_i w_i f_i}{\sum_i w_i} \tag{8}$$

Considering the backpropagation algorithm of the gradient descent method, the error signals can be obtained. This algorithm minimizes the training error by adjusting the customizable parameters.

*4.3. Genetic Algorithm*

The GA belongs to the meta-heuristic algorithm classification of evolutionary-based algorithms, which solve the problem with and without being constrained. It is based on the

natural selection of the evolutionary algorithms. Various optimization problems have led to the use of GA. It is a bio-inspired operator that makes use of mutation, crossover, as well as preference. Holland [25] suggested this technique depending on the natural selection procedure. Generating a randomly generated population of chromosomes is commonly the first stage in GAs. The fitness function is used to evaluate the extracted chromosomes. The chromosomes with a best approximation of optimal solution have a higher probability of reproducing. The GA parameters have a direct affect in the speed and the accuracy of the system regarding the appropriate convergence of the results. The appropriate crossover and mutation parameters guarantee the success rate of the GA [25]. The solution can be missed using a high mutation rate, which is close to the present state. On the other hand, the lower mutation rate can stick the process toward the local optimum. Furthermore, crossover prohibits offspring from being generated within the new generation; as a result, it will not be an identical replica of their parents' previous population. The mutation index was set to a fair value and the crossover parameter to a more significant value, as De Jong and Michelle also tried [26]. The GA-based method for fixture layout's machining was performed as: initialization of GA control parameters, fitness evaluation, new population (crossover, reproduction, mutation), and assessing the new population and termination criteria [27]. The GA results show that the crossover, mutation, generation, and population parameters have all been selected appropriately.

The right definition of the cost function is extremely important in optimization in order to reach highest possible system efficiency. In our study, the cost function is defined as follows:

$$J\,(t_1, t_2, t_3) = 10^4 \times \text{MSE(Actual} - \text{Predicted)} \tag{9}$$

where $t_1$, $t_2$, and $t_3$ are the cutting parameters including feed rate, cutting speed, and depth of cut, respectively. Actual and predicted are the specific tool wear or surface roughness captured via the experimental or calculated result via the machine learning method.

## 5. Results and Discussions

In this research, machining parameters have been modelled for the calculation of the tool wear and surface roughness based on the cutting parameters in the milling process. Then, the multi-objective GA function of MATLAB was used to extract the optimal cutting parameters of the process.

The MATLAB model used in Section 3's GA optimization approach was created. The model provided in Section 4.2 is used to propose ANFIS. The model is then fine-tuned using the GA optimization approach described in Section 4.3.

MATLAB's "ANFIS function" was employed to create the ANFIS simulation model. All of the findings in the Section 3 are illustrated using the "plot" function of MATLAB, which was used to create the algorithms. In addition, MATLAB's "ga" and "gamultiobj" functions were used to create the GA and multi-objective GA, respectively. The proposed ANFIS, comprising ANFIS-TW and ANFIS-SR, has optimum hyperparameters that may be extracted using the single-objective GA optimization approach. The suggested ANFIS-SR and ANFIS-TW optimized hyperparameters for the milling process to forecast tool wear and surface roughness are shown in Table 2.

**Table 2.** Hyperparameters of ANFIS for surface roughness and tool wear prediction.

| Model | MF | Epoch Number | Initial Step Size | Step Size Decrease Rate | Step Size Increase Rate |
|-------|-----|--------------|-------------------|-------------------------|-------------------------|
| SR | 3 | 404 | 0.09690 | 0.97532 | 1.14378 |
| TW | 4 | 435 | 0.06559 | 0.98900 | 1.36466 |

Figure 3 shows the optimization process of single-objective GA in the calculation of the optimum hyperparameters of the ANFIS-SR for maximum generation of 50. As is shown in Figure 3, the optimization is terminated after 44 generations as there was not any changed in the best value, which is the RMSE between the actual and predicted surface

roughness via the ANFIS model. The extracted optimal hyperparameters are reported in the second row of Table 2. The employed cost function inside the single objective GA is based on Equation (9).

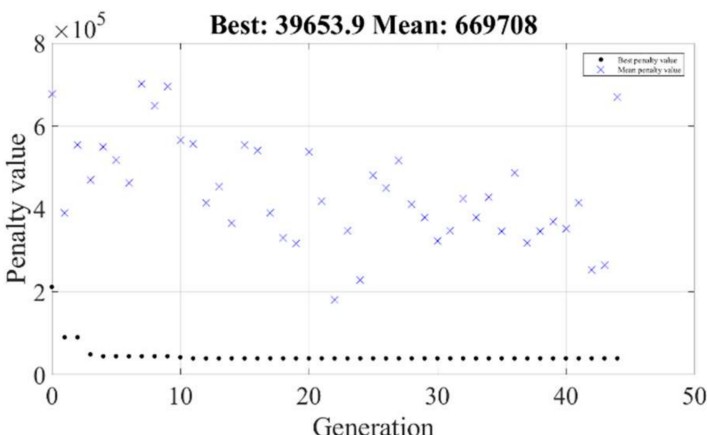

**Figure 3.** The convergence of single GA optimization method for ANFIS-SR.

Figure 4 shows the optimization process of single-objective GA in calculation of the optimized hyperparameters of the ANFIS-TW for maximum generation of 50. As is shown in Figure 4, the optimization is not terminated during the 50th generation. The extracted optimal hyperparameters are reported in the third row of Table 2. The employed cost function inside the single objective GA is based on Equation (9).

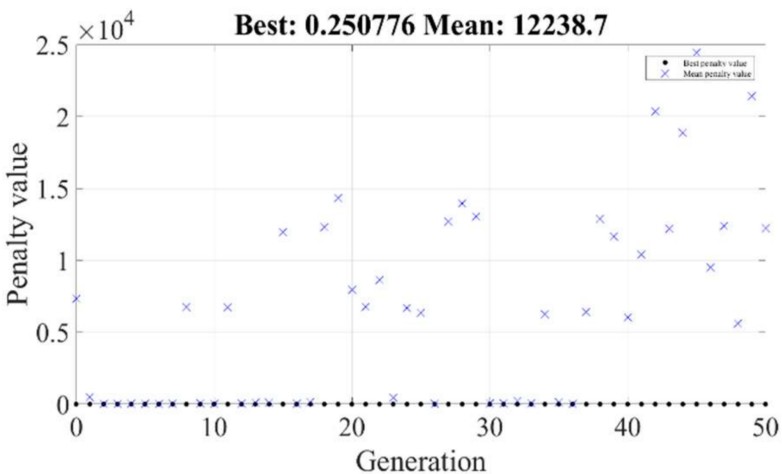

**Figure 4.** The convergence of single GA optimization method for ANFIS-TW.

Figure 5 shows the actual and predicted value of the specific tool wear and surface roughness. Seven datasets were chosen for training (78%), and two for the developed models testing (22%). According to the results illustrated in Figure 5a, it is quite apparent that ANFIS-SR predictions are more accurate compared to the ANFIS-TW during the testing and training procedures for the algorithms. It should be noted that the RMSE between the actual and predicted surface roughness of the milling process using the experimental setup and ANFIS-SR was $8.2086 \times 10^{-16}$ (mm) based on Figure 5a. In addition, Figure 5b shows that the RMSE between the experimentally captured and predicted via ANFIS-TW of specific tool wear was $1.0649 \times 10^{-15}$ (cm/(cm$^3$/s)). In addition, the MSE between the predicted and actual surface roughness and specific tool wear using the experiment, ANFIS-SR, and ANFIS-TW were $6.7382 \times 10^{-31}$ (mm) and $1.134 \times 10^{-30}$ (cm/(cm$^3$/s)) based on Figure 5, respectively.

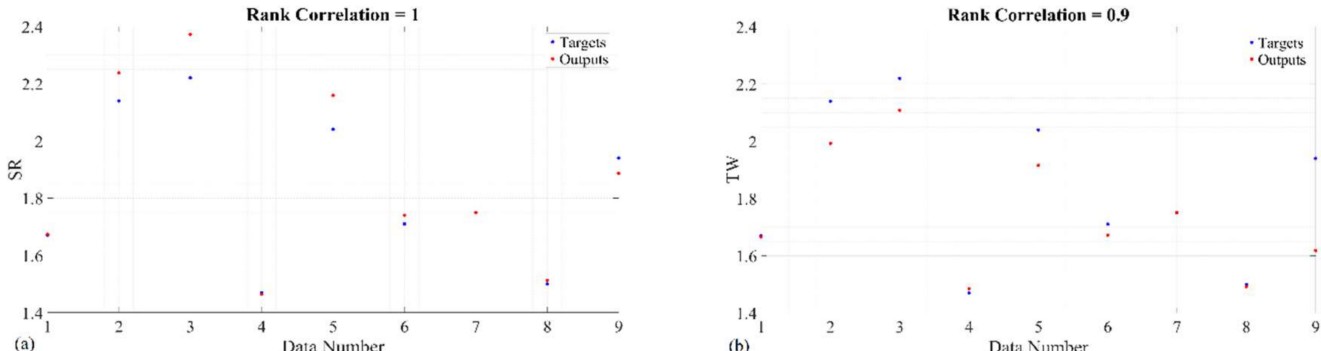

**Figure 5.** The actual and predicted outcomes during whole data including testing and training using (**a**) ANFIS-SR; (**b**) ANFIS-TW.

The *t*-test was carried out between the target and output values of surface roughness. The two-tailed *p* value was equal to 0.9323. By conventional criteria, this difference is considered to be not statistically significant. In addition, the standard error of difference is 0.135. Additionally, the *t*-test has been carried out between the target and output values of tool wear. The two-tailed P value was equal to 0.9167. The mean of Group One minus Group Two was equal to $-0.000322$. There was a 95% confidence interval of this difference (from $-0.006747$ to 0.006103). The standard error of difference equaled 0.003.

Figure 6 shows the histogram of error for the predicted data during the models' testing and training procedures, including ANFIS-SR and ANFIS-TW. The error histogram of ANFIS-SR is right-skewed according to Figure 6a.

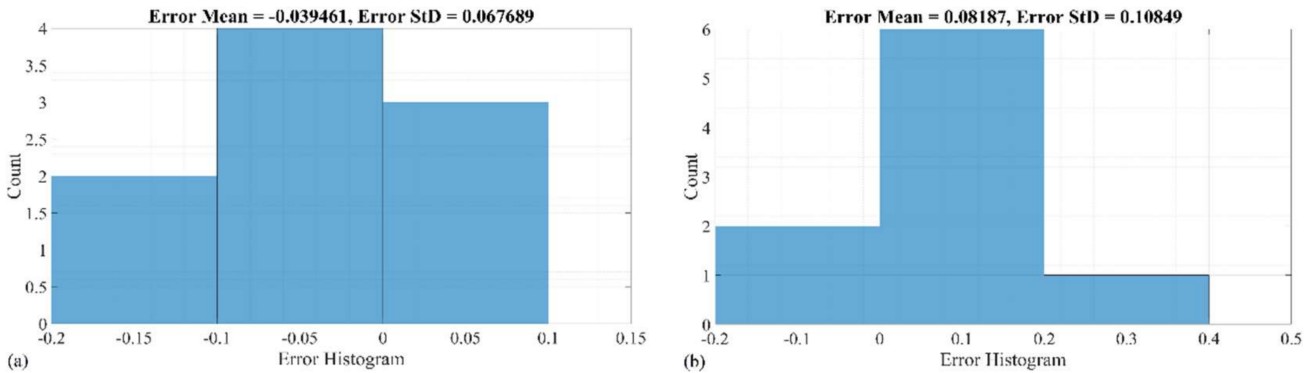

**Figure 6.** The error histogram of the data during the testing and training processes for (**a**) ANFIS-SR; (**b**) ANFIS-TW.

It has a smaller range of error that is between $-0.2$ and 0.1 (mm). In addition, the error histogram for ANFIS-TW is left-skewed in the range of $-0.2$ to 0.4 (cm/(cm$^3$/s)). The average errors between the predicted and actual surface roughness using ANFIS-SR and specific tool wear ANFIS-TW were $-3.9461 \times 10^{-2}$ (mm) and $8.187 \times 10^{-2}$ (cm/(cm$^3$/s)), respectively. In addition, the variation of error histogram using ANFIS-SR and ANFIS-TW were $6.7689 \times 10^{-2}$ (mm) and $1.0849 \times 10^{-1}$ (cm/(cm$^3$/s)), respectively.

The regression of the developed two methods, including ANFIS-SR and ANFIS-TW for whole data (i.e., testing and training), are, respectively, presented in Figure 7. ANFIS-SR and ANFIS-TW during training and testing were 0.96141 and 0.78718, respectively.

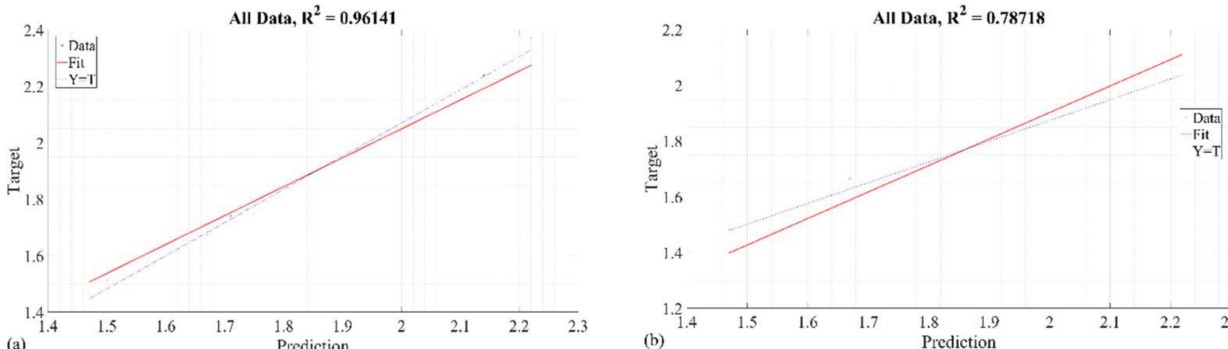

**Figure 7.** The regression of the data during the testing and training processes for (**a**) ANFIS-SR; (**b**) ANFIS-TW.

Based on a paired configuration of the inputs and outputs of ANFIS-SR, Figure 8 displays the derived fuzzy rule surface. Figure 8a depicts the effect of feed rate and cutting speed on output (surface finish). As seen in Figures 8 and 9, all parameters, inputs, and outputs have been standardized before being shown.

Figure 8b depicts the output variation as a function of feed rate and depth of cut. Finally, Figure 8c shows the change in surface polish as a function of cutting depth and cutting speed. Surface finish is greatly affected by feed rate, as seen in Figure 8b,c. Feed rate has a greater impact on surface finish than cutting speed, as seen in Figure 8a. On top of that, based on the data shown in Figure 8b, the feed rate has more of an effect than the cut depth does. The deeper the depth of cut, the better the surface finish will be as a consequence of the feed effect. Finally, Figure 8c demonstrates that the depth of cut has a greater influence on surface finish variation than velocity.

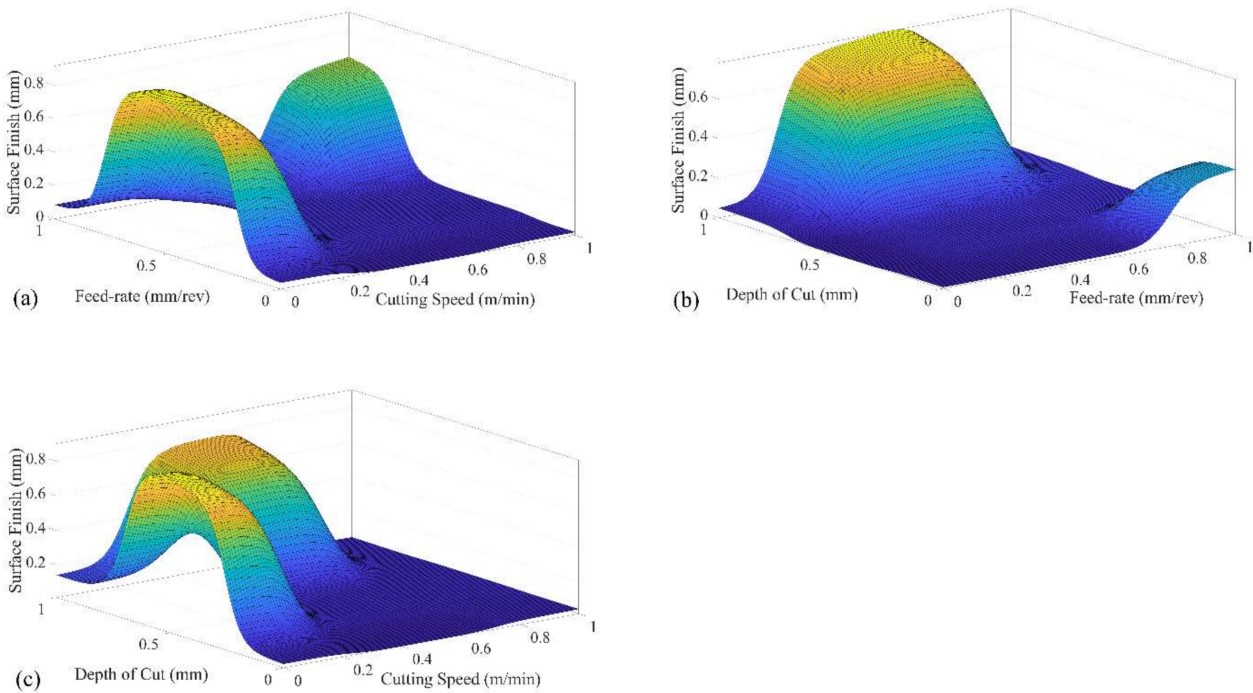

**Figure 8.** The ruled surface of ANFIS-1 (ANFIS-SR) in terms of cutting parameters (**a**) SR at feed-rate and cutting speed; (**b**) SR at depth of cut and feed-rate; (**c**) SR at depth of cut and cutting speed.

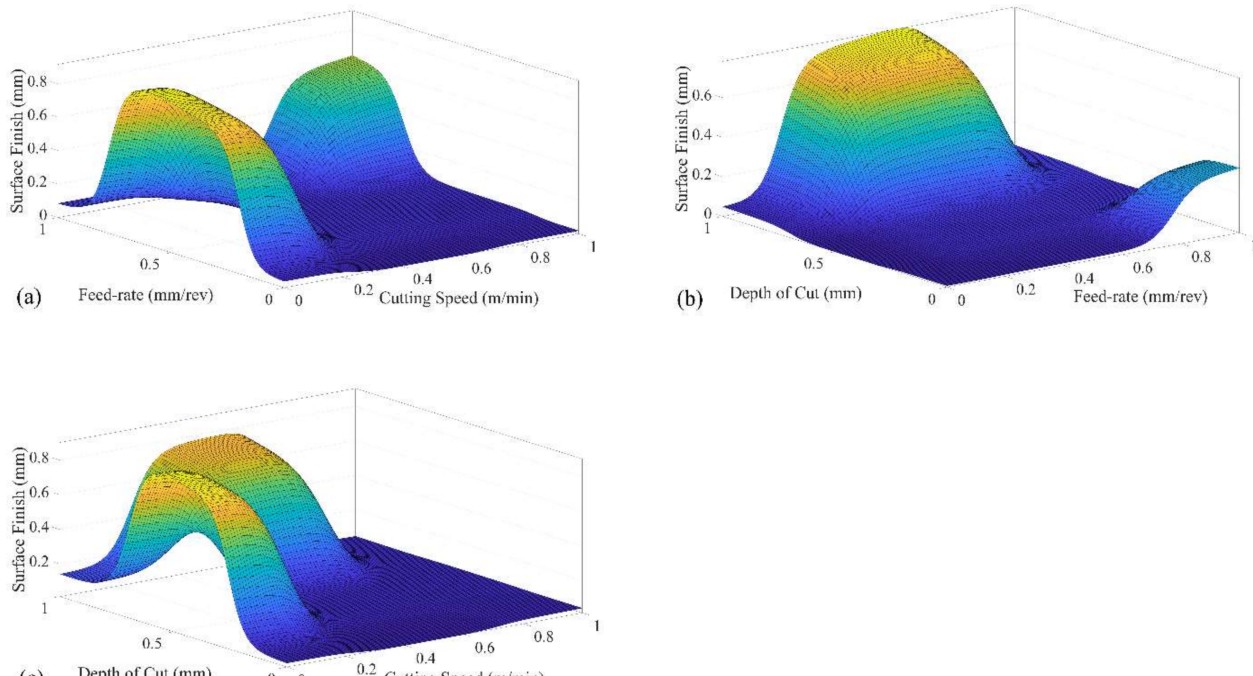

**Figure 9.** The ruled surface of ANFIS-2 (ANFIS-TW) in terms of cutting parameters (**a**) TW at feed-rate and cutting speed; (**b**) TW at depth of cut and feed-rate; (**c**) TW at depth of cut and cutting speed.

On the other hand, the fuzzy rule surface is shown in Figure 9. According to changes in feed rate and cutting speed, Figure 9a depicts the variance in output (tool wear). The parameters in this figure, including inputs and outputs, have been standardized. Figure 9b depicts the output's response to changes in feed rate and cut depth. Figure 9c illustrates how tool wear changes with cutting depth and cutting speed. The depth of cut has a significant impact on tool wear, as seen in Figure 9. Figure 9a shows that cutting speed has a greater impact on tool wear than feed rate. As shown in Figure 9b, the depth of cut has a greater effect on the feed rate than it does on the feed speed.

The multi-objective GA is employed using the extracted ANFIS-SR and ANFIS-TW models, while the predicted specific tool wear and surface roughness are the objective values of the method. The 14 optimal solutions are extracted, which are shown in Table 3.

**Table 3.** The obtained results using multi-objective GA.

|  | Cutting Speed (m/min) | Feed Rate (mm/rev·tooth) | Depth of Cut (mm) |
|---|---|---|---|
| 1 | 256.5 | 0.1005 | 1.2735 |
| 2 | 256.9 | 0.1388 | 1.2777 |
| 3 | 255.0 | 0.1424 | 1.3012 |
| 4 | 256.5 | 0.1023 | 1.2746 |
| 5 | 252.6 | 0.1431 | 1.3108 |
| 6 | 253.8 | 0.1421 | 1.2940 |
| 7 | 256.4 | 0.1396 | 1.2871 |
| 8 | 256.7 | 0.1396 | 1.2795 |
| 9 | 254.0 | 0.1410 | 1.2905 |
| 10 | 256.5 | 0.1005 | 1.2735 |
| 11 | 252.7 | 0.1429 | 1.3085 |
| 12 | 255.2 | 0.1396 | 1.2883 |
| 13 | 252.9 | 0.1427 | 1.3026 |
| 14 | 256.9 | 0.1388 | 1.2777 |

## 6. Conclusions

In this paper, a new ANFIS model for optimizing workpiece mill parameters was studied. The workpiece was AISI 1045 steel alloy block. The input parameters were three independent variables of cutting and feeding speeds as well as cutting depth. From a manufacturing point of view, the hardest part was the extraction of the data that needs a highly equipped lab to hold the experiments and measure relevant results. However, the main contribution of this study was to reduce this cost from further experiments via extracting the optimal milling process parameters. Therefore, the aim was optimizing surface roughness and special tools parameters that were considered output parameters. In the proposed method for extracting optimal ANFIS meta-parameters for the model, single-objective GA was used. The complexity of the proposed algorithm can be discussed from a different point of view. In the point of computing, ANFIS is one non-complex machine learning method that lower CPU power PCs can train. The multi-objective GA can extract the optimal results in less than 5 min. Here, the modeling tool was MATLAB software, and the results were extracted based on data segmentation for 78% of training and 22% of testing. By looking closely at the results, it can be seen that the ANFIS can achieve better results during testing and training. Then, the extracted optimal ANFIS models for calculation of tool wear and surface roughness are employed inside the multi-objective GA to extract the optimal cutting parameters to reach the best solution, which is the lowest thrust force as well as surface roughness.

**Author Contributions:** Conceptualization, S.P. (Siamak Pedrammehr) and M.H.; methodology, S.P. (Siamak Pedrammehr) and M.R.C.Q.; software, M.R.C.Q.; validation, S.P. (Sajjad Pakzad) and H.P.; formal analysis, S.P. (Siamak Pedrammehr), M.H. and M.R.C.Q.; investigation, S.P. (Siamak Pedrammehr), M.M.E. and M.R.C.Q.; resources, S.P. (Sajjad Pakzad) and H.P.; data curation, H.P. and M.M.E.; writing—original draft preparation, S.P. (Siamak Pedrammehr), and M.H.; writing—review and editing, S.P. (Siamak Pedrammehr), M.M.E. and A.H.S. visualization, S.P. (Sajjad Pakzad) and M.R.C.Q.; supervision, S.P. (Siamak Pedrammehr); project administration, M.M.E. All authors have read and agreed to the published version of the manuscript.

**Funding:** This research received no external funding.

**Institutional Review Board Statement:** Not Applicable.

**Informed Consent Statement:** Not Applicable.

**Data Availability Statement:** Not Applicable.

**Conflicts of Interest:** The authors declare there is no conflict of interest.

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
