# Peer review of "Machine Learning-Based Modelling and Meta-Heuristic-Based Optimization of Specific Tool Wear and Surface Roughness in the Milling Process"

_axioms, doi:10.3390/axioms11090430_

Round 1
Reviewer 1 Report
The purpose of this research is to investigate different milling parameters for optimization to achieve the maximum rate of material removal with the minimum tool wear and surface roughness. In this study a tool wear factor is specified to investigate tool wear parameters and amount of material removed during machining, simultaneously. The second output parameter is surface roughness. The DOE technique is used to design the experiments and applied to the milling machine. The practical data is used to develop different mathematical models. In addition, a single-objective genetic algorithm (GA) is applied to numerate the optimal hyperparameters of the proposed adaptive network-based fuzzy inference system (ANFIS) to achieve the best possible efficiency. Afterwards, the multi-objective GA is employed to extract the optimum cutting parameters to reach the specified tool wear and the least surface roughness. The proposed method is developed under MATLAB using the practically extracted dataset and neural network. This research study seems interesting and addresses timely subject in the field. Results are supported/justified scientifically and quality of presentation is considerably high. This manuscript could be accepted for publication after incorporation of comprehensive proofread throughout the manuscript to rectify the typo/grammatical errors.
Author Response
Response 1: The authors would like to appreciate the reviewer for the constructive comment and suggestions. According to the point, the authors proofread the paper and the grammar of the paper has also been rechecked.

Reviewer 2 Report
The authors should provide more details regarding the analysis of the results.
I suggest a significant re-write of the introduction. It should provide an overview of the importance of the main contribution of the proposed algorithm.
More up-to-date studies are suggested to be cited.
How to initialize the agents in ANFIS Algorithm?
It is necessary to discuss the complexity of the proposed solution.
Statistical analysis should be carried out to demonstrate that the experimental results are significant. Such as the ANOVA test and T-test
Author Response
The authors would like to appreciate the reviewer for the constructive comment and suggestions. Please check the attachment for all the revision parts.
Point 1: The authors should provide more details regarding the analysis of the results.
Response 1: The results and discussion sections are comprehensively modified to provide the T-test results based on the previous suggestion of Reviewer 2.
Point 2: I suggest a significant re-write of the introduction. It should provide an overview of the importance of the main contribution of the proposed algorithm.
Response 2: In order to follow the comment of Reviewer 2, the following explanation is added to the end of the introduction Section of the revised paper as follows:
“In this research, the specific tool wear is introduced as a new output parameter named in which material removal rate and tool wear are evaluated. The second target output is Surface roughness. Three alternative optimisation techniques have been put forward and contrasted to reduce tool wear, increase surface smoothness, and increase the material removal rate in an AISI 1045 alloy steel mill. The main contribution of this research is to extract the optimal milling process parameters, including cutting speed (m/min), feed rate (mm/rev⸱tooth), and depth of cut (mm), to reach the optimal specific tool wear (mm) and surface roughness (cm/(cm3/s)) using multi-objective GA. It should be noted that higher specific tool wear means lower surface roughness. There is a need for a Pareto front to extract the optimal solutions for the milling process. In addition, the multi-objective functions of the milling process are extracted using an adaptive network-based fuzzy inference system (ANFIS) to imitate the process based on the results of an experiment. The optimal solutions are recommended at the end of this study.”
Point 3: More up-to-date studies are suggested to be cited.
Response 3: Per the reviewers comment recently published research works which are related to use of machine learning in different predictions in milling operation have been added to the paper.
- Rajesh, A.S., Prabhuswamy, M.S., Krishnasamy, S., Smart Manufacturing through Machine Learning: A Review, Perspective, and Future Directions to the Machining Industry. J. Eng. 2022.
- Mohanraj, T., Yerchuru, J., Krishnan, H., Aravind, R.N., Yameni, R., Development of tool condition monitoring system in end milling process using wavelet features and Hoelder’s exponent with machine learning algorithms. Measurement, 2021, 173, 108671.
- Traini, E., Bruno, G. and Lombardi, F., Tool condition monitoring framework for predictive maintenance: a case study on milling process. Int. J. Prod. Res. 2021, 59, 7179-7193.
- Wang, R., Song, Q., Liu, Z., Ma, H., Gupta, M.K., Liu, Z., A novel unsupervised machine learning-based method for chatter detection in the milling of thin-walled parts. Sensors, 2021, 21, 5779.
- Yu, Y.Y., Zhang, D., Zhang, X.M., Peng, X.B., Ding, H., Online stability boundary drifting prediction in milling process: An incremental learning approach. Mech. Syst. Signal Process. 2022, 173, 109062.
- Charalampous, P., Prediction of cutting forces in milling using machine learning algorithms and finite element analysis. J. Mater. Eng. Perform. 2021, 30, 2002-2013.
- Li, R., Yao, Q., Xu, W., Li, J., Wang, X., Study of cutting power and power efficiency during straight-tooth cylindrical milling process of particle boards. Materials, 2022, 15, 879.
- Ramesh, P., Mani, K., Prediction of surface roughness using machine learning approach for abrasive waterjet milling of alumina ceramic. Int. J. Adv. Manuf. Technol, 2022, 119, 503-516.
- Uhlmann, E., Holznagel, T., Schehl, P., Bode, Y., Machine learning of surface layer property prediction for milling operations. J. manuf. mater. Process. 2021, 5, 104.
Point 4: How to initialise the agents in ANFIS Algorithm?
Response 4: Two agents are collaborating with our proposed ANFIS model inside the multi-objective GA. The first agent is responsible for the calculation of specific tool wear (mm) using cutting speed (m/min), feed rate (mm/rev⸱ tooth), and depth of cut (mm). Then, it should be trained using the cutting speed (m/min), feed rate (mm/rev⸱ tooth), and depth of cut (mm) as inputs and specific tool wear (mm) as outputs. On the other hand, the second agent is initiated using the same inputs as the first agent (including cutting speed (m/min), feed rate (mm/rev⸱ tooth), and depth of cut (mm)). At the same time, it extracted the surface roughness (cm/(cm3/s)).
Point 5: It is necessary to discuss the complexity of the proposed solution.
Response 5: The complexity of the proposed algorithm can be discussed from a different point of view. In point of computing, ANFIS is one non-complex machine learning method that lower CPU power PCs can train. However, after extraction of the machine learning-based model (ANFIS), it can recalculate the results in a few seconds. Then, the multi-objective GA can extract the optimal results in less than 5 minutes. In the point of manufacturing, the hardest part is the extraction of the data that needs a highly equipped lab to hold the experiments and measure relevant results. However, the main contribution of this study is to reduce this cost from further experiments via extracting the optimal milling process parameters.
Point 6: Statistical analysis should be carried out to demonstrate that the experimental results are significant. Such as the ANOVA test and T-test
Response 6: The T-Test has been held via https://www.graphpad.com/quickcalcs/ttest1.cfm, and the results for surface roughness are reported as below:
The results for tool wear are reported as below:
In addition, the following explanation is added inside the results and discussion section as follows:
“The T-test has been carried out between the target and output values of surface roughness. The two-tailed P value is equal to 0.9323. By conventional criteria, this difference is considered to be not statistically significant. In addition, standard error of difference is 0.135. Also, the T-test has been carried out between the target and output values of tool wear. The two-tailed P value is equal to 0.9167. The mean of Group One minus Group Two is equal to -0.000322. There is 95% confidence interval of this difference (from -0.006747 to 0.006103). The standard error of difference equals 0.003.”.

Round 2
Reviewer 2 Report
you can add a statistical figure, such as Q-Q Plot or Histogram
Read this paper; you can use
Alhussan, Amel Ali, Doaa Sami Khafaga, El-Sayed M. El-kenawy, Abdelhameed Ibrahim, Marwa M. Eid, and Abdelaziz A. Abdelhamid. "Pothole and Plain Road Classification Using Adaptive Mutation Dipper Throated Optimization and Transfer Learning for Self Driving Cars." IEEE Access (2022).